# Efficiently correcting patch-based segmentation errors to control image-level performance in retinal images

**Patrick Köhler**[1]                     PATRICK.KOEHLER@UNI-TUEBINGEN.DE
**Jeremiah Fadugba**[2,3]             JFADUGBA@QUANTUMLEAPAFRICA.ORG
**Philipp Berens**[1,4]                PHILIPP.BERENS@UNI-TUEBINGEN.DE
**Lisa M. Koch**[1,5]                  LISA.KOCH@UNI-TUEBINGEN.DE

[1] *Hertie Institute for AI in Brain Health, University of Tübingen, Germany*

[2] *African Institute for Mathematical Sciences (AIMS), Rwanda*

[3] *University of Ibadan, Nigeria*

[4] *Tübingen AI Center, University of Tübingen, Germany*

[5] *Department of Diabetes, Endocrinology, Nutritional Medicine and Metabolism UDEM, Inselspital, Bern University Hospital, University of Bern, Switzerland*

**Editors:** Accepted for publication at MIDL 2024

## Abstract

Segmentation models which are deployed into clinical practice need to meet a quality standard for each image. Even when models perform well on average, they may fail at segmenting individual images with a sufficiently high quality. We propose a combined quality control and error correction framework to reach the desired segmentation quality in each image. Our framework recommends the necessary number of local patches for manual review and estimates the impact of the intervention on the Dice Score of the corrected segmentation. This allows to trade off segmentation quality against time invested into manual review. We select the patches based on uncertainty maps obtained from an ensemble of segmentation models. We evaluated our method on retinal vessel segmentation on fundus images, where the Dice Score increased substantially after reviewing only a few patches. Our method accurately estimated the review's impact on the Dice Score and we found that our framework controls the quality standard *efficiently*, i.e. reviewing as little as necessary.

**Keywords:** Quality Control, Segmentation, Retinal Blood Vessels, Fundus

## 1. Introduction

Segmentation is a central task in medical image analysis, as it often builds the foundation for surgical planning (Li et al., 2021), diagnosis and disease progression monitoring (Soomro et al., 2019). In ophthalmology for example, segmenting retinal blood vessels from fundus images provides geometric characteristics such as branching angles or vessel diameters in a non invasive fashion. Unfortunately, their manual segmentation requires three to five hours per image (Jin et al., 2022), making it unfeasible to annotate entire images routinely for every patient. Recently, medical image segmentation algorithms have achieved a performance that is sufficient for clinical deployment (Isensee et al., 2021). Yet, even the best models are not guaranteed to perform well on *all* images and may fail silently on individual ones.

In medical contexts, quality standards are often crucial for safety, fairness and efficacy of therapeutic decisions. One strategy to implement such standards is to predict a

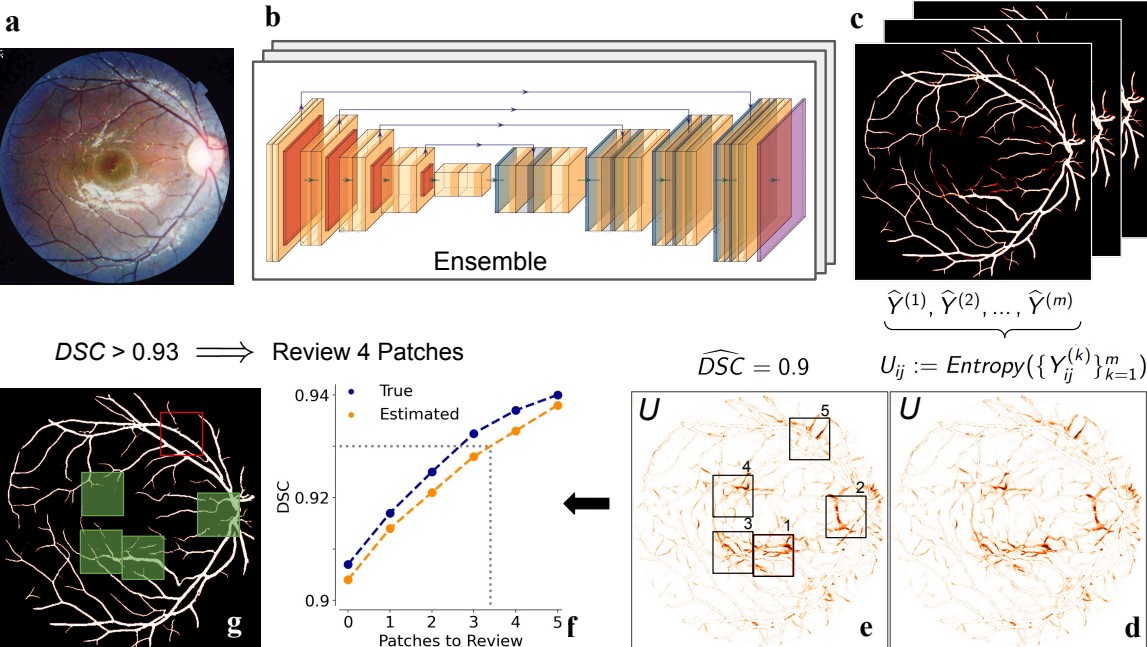

Figure 1: Overview. We obtain multiple predictions from an ensemble (**a-c**) and compute an uncertainty map (**d**) from which we select patches with highest uncertainty (**e**). Afterwards, we estimate the DSC and how it would change if the patches were to be reviewed manually (**f**) to satisfy a target quality (**g**).

quality metric such as the Dice Score Coefficient (DSC) per image without knowing the ground truth and exclude low quality segmentations from downstream analyses. This can be done by training an auxiliary DSC regression network in addition to the segmentation model itself (Robinson et al., 2018; Williams et al., 2021; Fournel et al., 2021), or using probabilistic segmentation model outputs (Li et al., 2022). Similarly, Galdran et al. (2018) learn to predict the normalized mutual information between the ground truth (GT) and the model segmentation, leveraging manually degraded GTs as a training set. However, this quality control paradigm defers entire images with subpar DSC for manual review. This may cause more manual labour than necessary because the low performance may be caused predominantly by specific image regions. For a quality assessment which is more granular than image level, Zaman et al. (2023) train a model to predict segmentation error maps. While such approaches identify segmentation errors they do not discuss effective strategies for corrective interventions. This is done by interactive segmentation methods (Liu et al., 2022a; Luo et al., 2021), which incorporate manual annotations to refine their prediction. Similarly, Benenson et al. (2019) propose an error correction mechanism by training an additional network to correct the predicted segmentation. This approach relies on a manually curated dataset of corrections which is representative for the model's failure modes. Neither of these methods has attempted to quantify the effect of the correction in advance.

In contrast, here we propose a combined quality control and error correction framework for vessel segmentation in retinal fundus images. Our framework proposes local candidate regions for manual review instead of deferring entire images. Then, we extend a recently proposed DSC estimator to provide an estimate of the correction's impact (Li et al., 2022). This allows the annotator to assess how many patches should be reannotated in order to meet the required segmentation quality standards.

## 2. Methods

### 2.1. Multi-disease dataset for challenging vessel segmentation

We used the FIVES dataset (Jin et al., 2022), comprised of 800 high-resolution retinal fundus images from the Second Affiliated Hospital of Zhejiang University (SAHZU), China. The images were taken from healthy individuals and patients with glaucoma, age-related macular degeneration (AMD) and diabetic retinopathy (DR) (200 images each). In each image, retinal blood vessels were annotated manually by two junior annotators and verified by experienced senior annotators in a standardized procedure (Jin et al., 2022). The presence of disease lesions made the segmentation task more challenging because these could interfere with the blood vessels. We used the original splits provided with the dataset, i.e. 600 training images, from which we used 120 for validation, and 200 test images. All images were pre-processed by applying Contrast Limited Histogram Equalization (Pizer et al., 1987) with a clip limit of 2 and a grid size of 8 x 8. The images and segmentation masks with original resolution of 2048 x 2048 were resampled to 512 x 512 pixels.

### 2.2. Probabilistic segmentation model for vessel segmentation

We used the state-of-the-art vessel segmentation model FR-Unet (Liu et al., 2022b) to develop our framework. The architecture was optimized for the intricacies of retinal vessel segmentation, namely thin foreground structures and low-contrast regions. The model's hidden representations expanded horizontally and vertically through a multiresolution convolution mechanism to retain the full image resolution. This allowed aggregating features from different scales to supplement high-level contextual information to the low-level regimes and vice versa. In experiments with a limited range of common datasets, FR-Unet has been shown to outperform other architectures with fewer parameters (Liu et al., 2022b).

We used an ensemble of $m$ FR-Unets, which were trained with different random seeds (Ganaie et al., 2022). An image of size $n \times n$ was passed through the ensemble, resulting in $m$ predicted probabilistic segmentations $\widehat{Y}_p^{(1)}, \ldots, \widehat{Y}_p^{(m)}$ (Fig. 1 **a-c**). The final probabilistic segmentation $\widehat{Y}_p = \{p_i : p_i \in [0,1], i = 1, \ldots, n^2\}$ was obtained by averaging the individual outputs. Thresholding $\widehat{Y}_p$ yielded the predicted binary segmentation $\widehat{Y} = \mathbf{1}_{[\widehat{Y}_p > 0.5]}$.

### 2.3. Quality control framework

Given a fundus image and a probabilistic segmentation, our goal was to refer a minimal number of patches to manual review such that a desired segmentation quality could be guaranteed (Fig. 1). Our proposed framework consisted of two major components: (1) We computed pixel-wise uncertainties from the outputs of the FR-Unet ensemble to select

patches as candidates for manual review (Fig. 1 **a**-**e**). (2) We estimated each patch's impact on the segmentation quality if it was to be reviewed by an expert (Fig. 1 **f**, **g**). This allowed us to control the desired segmentation performance efficiently, i.e. only referring as few patches for review as necessary.

### 2.3.1. SELECTING CANDIDATE PATCHES FOR MANUAL SEGMENTATION

To obtain an uncertainty map $U$, we computed the pixelwise entropy across all $i = 1, \ldots, m$ probabilistic segmentations $\widehat{Y}_p^{(i)}$ (Fig. 1 **d**) We expected that high uncertainty regions correspond to erroneous segmentations and that reviewing those would lead to efficient quality improvements. To identify local regions with potentially low segmentation quality and therefore high potential for improvement, we convolved $U$ with a square kernel of size $P \times P$, effectively computing the mean uncertainty in the patch around each pixel. We then selected the $k$ non-overlapping patches with the highest uncertainty (Fig. 1 **e**).

### 2.3.2. ESTIMATING SEGMENTATION QUALITY WITHOUT GROUND-TRUTH LABELS

Having identified candidate patches, we wanted to estimate their impact on the image's segmentation quality in terms of DSC if they were to be re-segmented manually. The DSC is defined as

$$\text{DSC} = \frac{2\text{TP}}{2\text{TP} + \text{FP} + \text{FN}} = \frac{2\text{TP}}{(\text{TP} + \text{FN}) + (\text{TP} + \text{FP})} \;, \tag{1}$$

where, TP, FP and FN denote the number of true/ false positive and false negative pixels. Therefore, the DSC can only be computed using the GT segmentation. As GT labels are not available at test time, we can not compute Eq. 1 directly to assess the impact of reannotating a given patch. However, DSC can be estimated only having access to the probabilistic model output $\widehat{Y}_p$ (Li et al., 2022). This approach relies on calibrated output probabilities, which means that for all $\pi \in [0, 1]$ exactly $\pi \cdot 100\%$ of the pixels with predicted probability $\pi$ actually belong to the foreground, such that the predicted probabilities reflect the correctness of the prediction accurately.

If the outputs were perfectly calibrated, summing over them would yield the number of pixels that belonged to the GT foreground (by definition). Li et al. (2022) leverage this property to construct their estimator

$$\widehat{\text{DSC}}(\widehat{Y}_p) = \frac{2 \sum_{i=1}^{n} \mathbf{1}_{[p_i > 0.5]} p_i}{\sum_{i=1}^{n} p_i + \sum_{i=1}^{n} \mathbf{1}_{[p_i > 0.5]}} \;, \tag{2}$$

where $\mathbf{1}_{[\cdot]}$ denotes the indicator function. Hence, summing over the output probabilities of all pixels that were classified as foreground yields an estimator for TP (enumerator of Eq. 1). Analogously, we can estimate the total number of GT foreground pixels, i.e. TP + FN, by summing over all output probabilities (Eq. 2, first summand in denominator). We applied temperature scaling (TS) to the model outputs as in Li et al. (2022) to calibrate $p_i$.

### 2.3.3. ESTIMATING QUALITY IMPROVEMENT AFTER MANUAL REVIEW

The DSC estimator provided a quality assessment for an individual image. While this may be useful for quality control in itself, it does not provide actionable insight for estimating the

effect of patch-based error correction. Here, we were interested in the following question: *What would the DSC be if an expert reviewed specific high-uncertainty patches* (Fig. 1 **f**)?

Therefore, we propose an estimator for the DSC of the corrected segmentation $\widehat{Y}_{\text{corr}}$ that is composed of two image parts: The high-confidence regions where the model output is accepted $\widehat{Y}_{\text{model}}$, and the manually reviewed patches $\widehat{Y}_{\text{manual}}$ in the high-uncertainty regions. The DSC of the combined segmentation $\widehat{Y}_{\text{corr}}$ can be expressed as as a linear combination of the DSC estimates of its components:

$$\widehat{\text{DSC}}(\widehat{Y}_{p,\,\text{corr}}) = w_{\text{model}}\widehat{\text{DSC}}(\widehat{Y}_{p,\,\text{model}}) + w_{\text{manual}}\widehat{\text{DSC}}(\widehat{Y}_{p,\,\text{manual}}) \ . \tag{3}$$

The DSC for $\widehat{Y}_{\text{model}}$ can be estimated with Eq. 2 by considering the foreground probabilities $p_i$ only for regions where the model output was accepted. For $\widehat{Y}_{\text{manual}}$, we assumed perfect manual segmentation performance for simplicity, i.e. $\widehat{\text{DSC}}(\widehat{Y}_{\text{manual}}) = 1$. The weights $w_i$ correspond to the fraction of predicted foreground within $\widehat{Y}_i$. We give a theoretical justification for this choice in App. A.

## 3. Results

### 3.1. Segmentation performance before error correction

We trained an ensemble of $m = 5$ FR-Unets on the training fold of the FIVES dataset (see Sec. 2.2) for 80 epochs with the DSC Binary Cross Entropy loss[1]. This loss was a combination of the softDSC loss (Milletari et al., 2016) with Cross-Entropy and has been observed to produce better generalization than softDSC alone (Liu et al., 2021; Ma et al., 2021; Galdran et al., 2022). To improve generalization, we augmented the data with random flips and rotations. We used the Adam optimizer with a learning rate of $10^{-4}$ and a cosine annealing scheduler (number of iterations $\leq 40$).

Afterwards, we selected the model with the highest validation DSC, leading to an average DSC of $0.887\pm0.094$ (mean $\pm$ SD) on the test set ($n = 200$). For 10% of the images, the DSC was below $0.828$ or above $0.934$ (Fig. 2 **a**). We found that many of the segmentation errors occured in the finer vessel structures and we mainly observed discontinuities of vessels and missing segments (see. Fig 5 in App. B). The uncertainty maps provided by the FR-Unet ensemble accurately identified segmentation errors in the images (see Fig 5 in App. B).

### 3.2. Correcting high-uncertainty regions increases segmentation quality

Manually segmenting high-uncertainty regions increased the segmentation quality (Fig. 2 **b**). Here, we selected the top $k \in \{0,\ldots,5\}$ non-overlapping patches of size $81 \times 81$ in the uncertainty map (as in Fig. 1 **e**) and replaced the predicted output with the GT to model a "perfect" human annotator. The impact of patch size is discussed in App. C.

We observed a gradual increase in DSC when correcting more patches. For example, the median DSC across all images increased by more than 0.02 after correcting five patches. This was much more than could be achieved by a simple baseline approach of selecting random patches (from inside the retinal fundus) instead of high entropy patches, which increased segmentation quality at a lower rate (grey points in Fig. 2).

---

1. Trained models and code: github.com/berenslab/MIDL24-segmentation_quality_control.

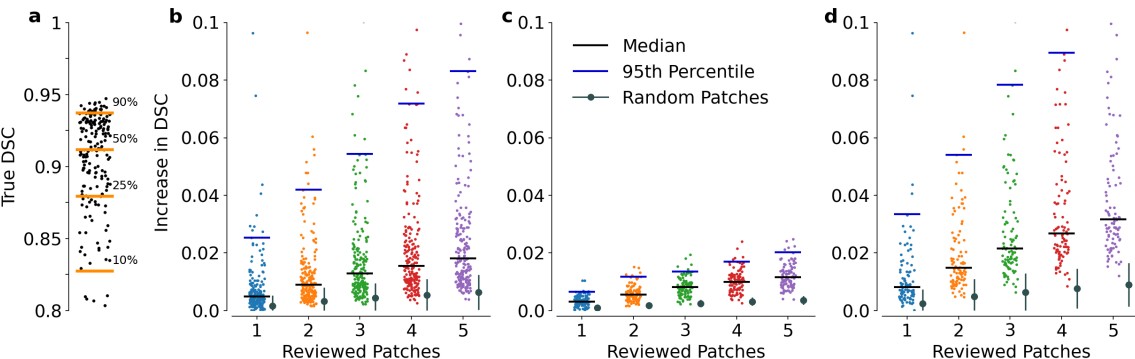

Figure 2: Distribution over DSC in the test set (**a**) and effect of reviewing the patches (**b-d**). For the selected patches, we inserted the GT to model manual correction. For the baseline we choose random patches and depict only mean, std for visual clarity. We split up the test set (**b**) into well (**c**, above median DSC) and poorly segmented images (**d**).

For further examination, we split the test set into well-segmented images (above median DSC, Fig. 2 **c**) and poorly segmented images (below median DSC, Fig. 2 **d**). The segmentation quality improved more for images which were initially poorly segmented (Fig. 2 **d**, similar trends were observed in a subgroup analysis in App. D). Here, we observed a median increase in DSC of approximately 0.04 when reannotating $k = 5$ patches. For some images, the segmentation quality was improved by up to 0.1. Even for images that were already well segmented the DSC could be further improved by 0.018 on average.

In summary, patch-wise error correction based on uncertainty led to a substantial increase in performance on average. However, using this simple patch selection strategy did not by itself provide an a-priori estimate for the effect of error correction.

### 3.3. DSC estimation allows to predict the impact of manual patch review

Therefore, we next predicted the effect of patch-wise error correction on the resulting segmentation quality, without access to the GT. We first assessed the estimator proposed by Li et al. (2022) (Eq. 2) to predict DSC before error correction. With a mean absolute error of 0.02, we found that it reliably estimated the segmentation quality for individual images (Fig. 3 **a**, red points), even though we observed a slight bias towards overestimating the true performance (mean error $\widehat{DSC} - DSC = 0.012$). Moreover, accurate DSC estimation relied heavily on calibration with temperature scaling. For uncalibrated model outputs, $\widehat{DSC}$ overestimated consistently (grey points in Fig. 3 **a**).

Ultimately, we were interested in predicting how the DSC changed if the segmentation in high uncertainty regions of the images was corrected. Hence, we evaluated our proposed DSC estimator for a patch-wise corrected segmentation (Eq. 3) and observed constantly low estimation errors across all number of patches (Fig. 3 **b**). The bias towards overestimation observed in the estimator from Li et al. (2022) carried over to our correction estimate. When we correct only one or two patches, the DSC's are being overestimated for more than half of

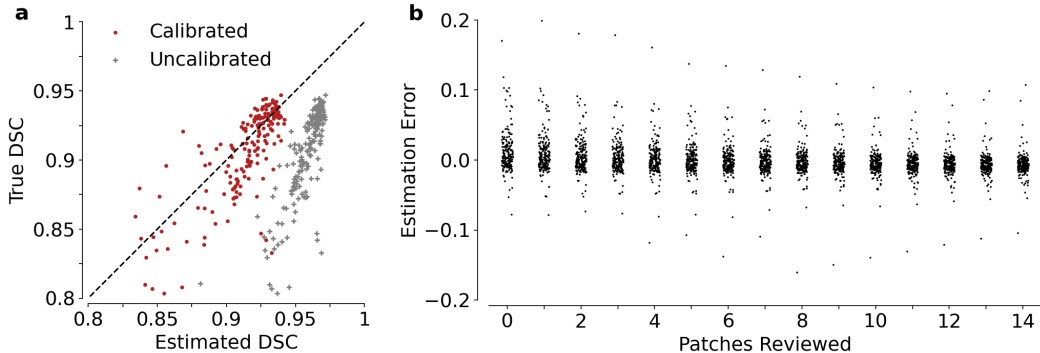

Figure 3: Accuracy of estimating DSC for test set images entirely segmented (**a**) by the model and (**b**) after manual review. Estimation error computed as $\widehat{\mathrm{DSC}} - \mathrm{DSC}$.

the images. In general, we preferred conservative estimates over those liberal ones, because we rather wanted to ensure the desired quality target with a higher probability at the cost of reviewing more patches. Hence, we introduced a correction term in the subsequent analysis to reduce overconfidence post-hoc.

In conclusion, the estimator from Li et al. (2022) was not only useful for its initial purpose of estimating average performance over entire datasets but also allowed us to assess how DSC would change if an expert was to review a specific set of patches.

### 3.4. Adaptive patch selection leads to more efficient resource allocation

Reviewing segmentations with human experts is typically time intensive and costly. Hence, we wanted to ensure that our method is efficient, i.e. that we only request as little human resources as necessary to reach the desired quality.

To quantify robustly the quality standard that has been reached by a correction strategy we chose the $5^{\mathrm{th}}$ percentiles over the images' DSCs. In contrast to the minimum DSCs, this accounted for outliers for which the model's segmentation is fundamentally incorrect and cannot be fixed with partial review. In our framework, the remaining 5% should be referred to full review or repeated acquisition. We calibrated our DSC estimates to prevent overconfidence (cf. Sec. 3.3) by subtracting $\epsilon = 0.02$. $\epsilon$ was optimized on the validation set such that the $5^{\mathrm{th}}$ percentiles matched the quality target.

We determined for each image individually how many patches needed to be reviewed in order to reach each of the three quality targets of DSC = 0.88, 0.90 and 0.92, where 0.92 was reported as human performance on this data set (Jin et al., 2022). With Eq. 3, we calculated the estimated DSC after reviewing $k = 1, \ldots, 14$ patches and select the lowest number of patches such that the estimated DSC exceeded the target. As a baseline, we chose a fixed number of patches which were reviewed for *each* image.

Our adaptive strategy led to 3.2 reviewed patches per image on average to reach a quality standard of almost 0.90 (Fig. 4). The actual performance ($5^{\mathrm{th}}$ percentiles were 0.87, 0.89, 0.91) differed slightly from the desired quality standards because of imperfect DSC estimation. For the baseline, approximately three times more patches were reviewed per image to achieve a similar quality standard.

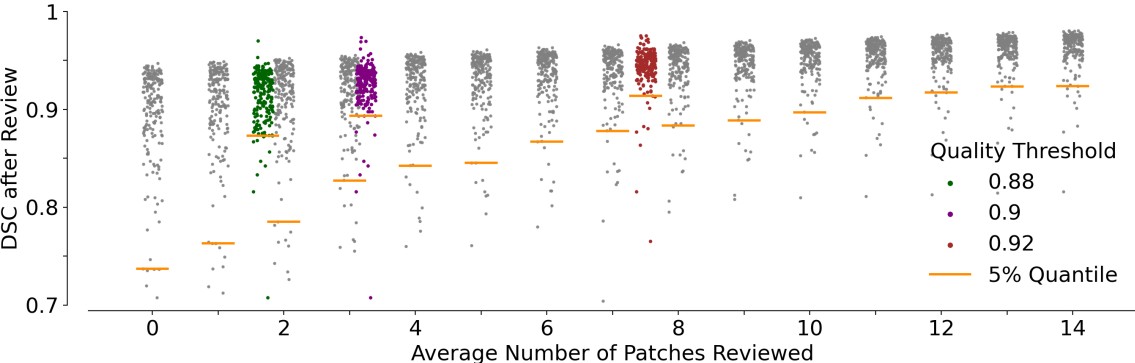

Figure 4: Efficiency comparison of our proposed workflow (coloured) with a baseline, where for each image an equal number of patches is reviewed (grey). For our workflow, we chose three quality targets and estimated for each image individually how many patches needed to be reviewed to achieve the targets.

## 4. Discussion and Conclusion

We presented a framework to correct segmentation errors locally in order to control the segmentation quality per image. Correcting patches with high uncertainty led to an increase in segmentation quality for retinal blood vessels. Furthermore, we could accurately predict this increase using a DSC estimator which did not require access to the GT segmentation. Therefore, our workflow allowed to allocate more resources to images with poor segmentations and not waste resources where outputs already satisfied the quality criterion.

Our method estimates DSC, which is a measure of overlap with broad applicability in many segmentation tasks. However, particularly to account for topological consistency in thin vessel structures, predicting customized metrics such as the centerline DSC (Shit et al., 2021) is an important next step for certain downstream tasks.

In this paper, we assumed perfect expert performance for the patch review. While the performance may be higher than in large scale annotation settings because the reviewer can focus their attention on few small areas, this simplification ignores potential inter-rater variability. One mitigation would be to evaluate our method with multi-annotator data, where the oracle patch could be provided by a different annotator.

We used deep ensembles to quantify uncertainty and suggest candidate patches for manual review. Our modular framework allows replacing this step with any other approach that generates uncertainty maps such as Monte-Carlo dropout (see Fuchs et al. (2022) for an overview and App. E for a comparison of uncertainty maps). As uncertainty estimation typically generates computational overhead, other strategies could be pursued to suggest candidate patches purely based on image statistics, e.g. by identifying low-contrast regions.

Beyond covering large demands in hospitals more efficiently, our method can be applied in clinical trials, when automatic volumetric measurements inform on effect sizes of drugs. In that case, our framework could improve the efficiency of drug development by ensuring that each volumetric estimate is accurate enough for the downstream analysis of interest.

## Acknowledgments

This work was supported by the German Science Foundation (BE5601/8-1 and the Excellence Cluster 2064 "Machine Learning — New Perspectives for Science", project number 390727645), the Carl Zeiss Foundation in the project "Certification and Foundations of Safe Machine Learning Systems in Healthcare", the Hertie Foundation and the Carnegie Corporation of New York (provided through the African Institute for Mathematical Sciences).

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

## Appendix A. Justification for the choice of $w_i$ in Equation 3

Let us assume we are given a predicted segmentation $X$ and the respective ground truth $Y$. Given two disjoint subsets of those, which we denote with subscripts, we would like to combine their individual DSC linearly to obtain the overall $\mathrm{DSC}(X, Y)$. Let us consider the DSC definition in terms of set sizes

$$\mathrm{DSC}(X,Y) = \frac{2|X \cap Y|}{|X| + |Y|} \tag{4}$$

$$\stackrel{!}{=} w_1 \mathrm{DSC}(X_1, Y_1) + w_2 \mathrm{DSC}(X_2, Y_2) \ , \tag{5}$$

and solve for $w_i$.

Furthermore, let the subsets be disjoint and their union the entire image

$$Z_1 \cup Z_2 = Z \ , \quad Z_1 \cap Z_2 = \varnothing \qquad \forall Z \in \{X, Y\}. \tag{6}$$

Multiplying each of the summands in Eq. 5 with 1 allows us to write them over the same denominator as on the right side of Eq. 4. For $i = 1$ set $j = 2$ and vice versa. Then

$$w_i \mathrm{DSC}(X_i, Y_i) = w_i \mathrm{DSC}(X_i, Y_i) \frac{\frac{|X_1|+|X_2|+|Y_1|+|Y_2|}{|X_i|+|Y_i|}}{\frac{|X_1|+|X_2|+|Y_1|+|Y_2|}{|X_i|+|Y_i|}} = w_i \frac{2|X_i \cap Y_i| \left(1 + \frac{|X_j|+|Y_j|}{|X_i|+|Y_i|}\right)}{|X_1| + |X_2| + |Y_1| + |Y_2|}, \tag{7}$$

where the denominator of the right side equals to $|X| + |Y|$ because of the prerequisits 6. Now, let us choose $w_i$ such that the enumerator of Eq. 4 equals the enumerator of Eq. 5 after Eq. 7 as been applied, i.e.

$$\sum_{i=1}^{2} 2w_i |X_i \cap Y_i| \left(1 + \frac{|X_j| + |Y_j|}{|X_i| + |Y_i|}\right) = 2|X \cap Y|. \tag{8}$$

Since

$$\sum_{i=1}^{2} 2|X_i \cap Y_i| = 2|X \cap Y| \ , \tag{9}$$

because of the prerequisites 6, solving Eq. 8 for $w_i$ yields

$$w_i^* = \left(1 + \frac{|X_j| + |Y_j|}{|X_i| + |Y_i|}\right)^{-1} = \frac{|X_i| + |Y_i|}{|X| + |Y|}. \tag{10}$$

In words, this is the sum of predicted foreground and GT foreground within the subset $i$ over the sum of predicted foreground and GT foreground in the entire image. As we have no access to the amount of GT foreground we approximate it with the predicted amount of foreground. Hence, we approximate $w_i$ with the number of predicted foreground pixels within the subset $i$ over the total number of predicted foreground pixels.

## Appendix B.  Example Images

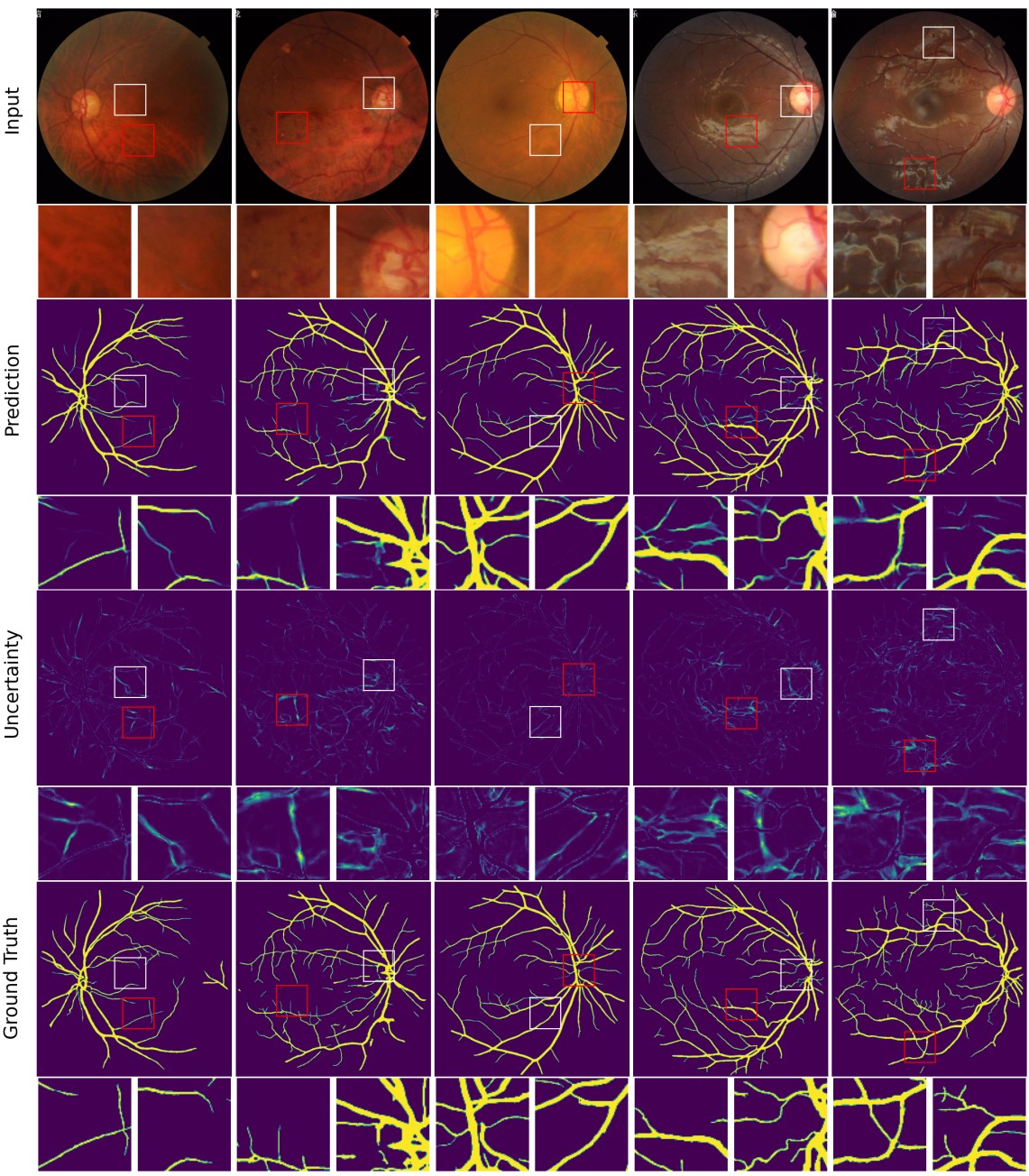

Figure 5: Example images drawn from the test set including the first two patches that were automatically selected as candidates for manual review (red indicates the first patch).

## Appendix C. Impact of patchsize

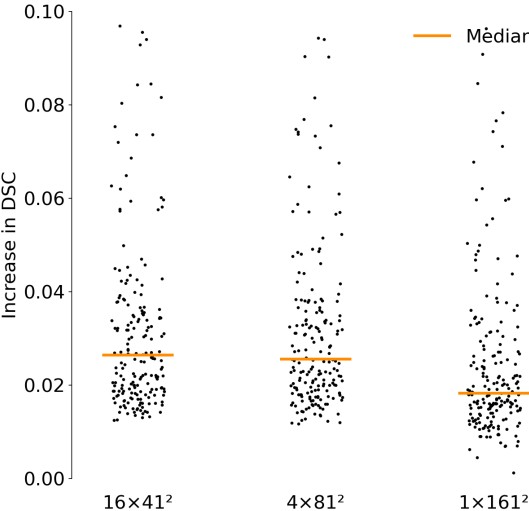

Figure 6: Impact of patchsize on the change in DSC. For the comparison of three patch sizes $(41^2, 81^2, 161^2)$, we choose the number of patches such that all three settings select the same area of the image, i.e. 1 patch of size $161^2$ vs 4 patches of size $81^2$ vs 16 times $41^2$. The effect strength is comparable between 16 small patches and 4 medium sized patches. We suppose that it is more convenient for a clinician to review fewer patches because delineating at the border of the patch requires special attention. Hence, we opt to show the results for patch size $81^2$ in the main text.

## Appendix D. Subgroup analysis

| AMD | DR | Glaucoma | Normal |
|---|---|---|---|
| $0.91 \pm 0.04$ | $0.89 \pm 0.05$ | $0.84 \pm 0.16$ | $0.90 \pm 0.04$ |

Table 1: Average $\pm std$ DSC per subgroup on the test set. Each subgroup consists of 50 images.

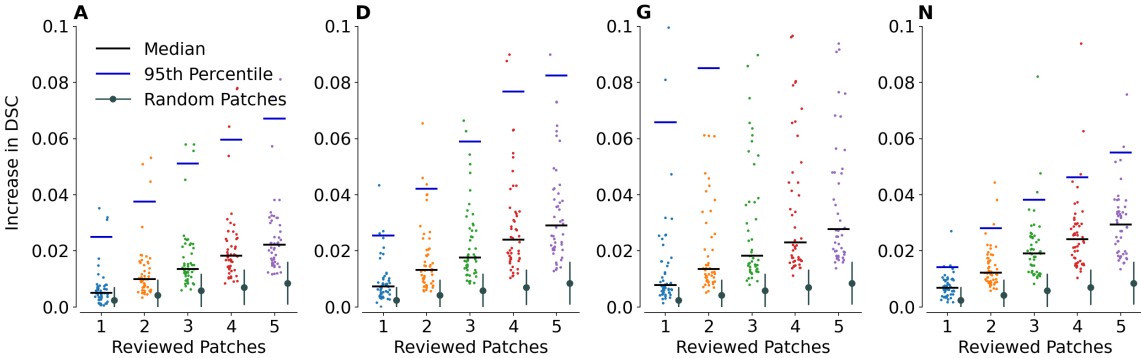

Figure 7: Effect of reviewing a specified number of patches split up by subgroup. A = Age Related Macular Degeneration, D = Diabetic Retinopathy, G = Glaucoma, N = Normal. Within the pathological subgroups, there exist more images that benefitted strongly from the review, whereas the effect size was distributed more uniformly in the subgroup of normal images. The effect was particularly pronounced for the Glaucoma subgroup, which had the lowest segmentation performance (Tab. 1).

## Appendix E. Comparison of uncertainty estimates

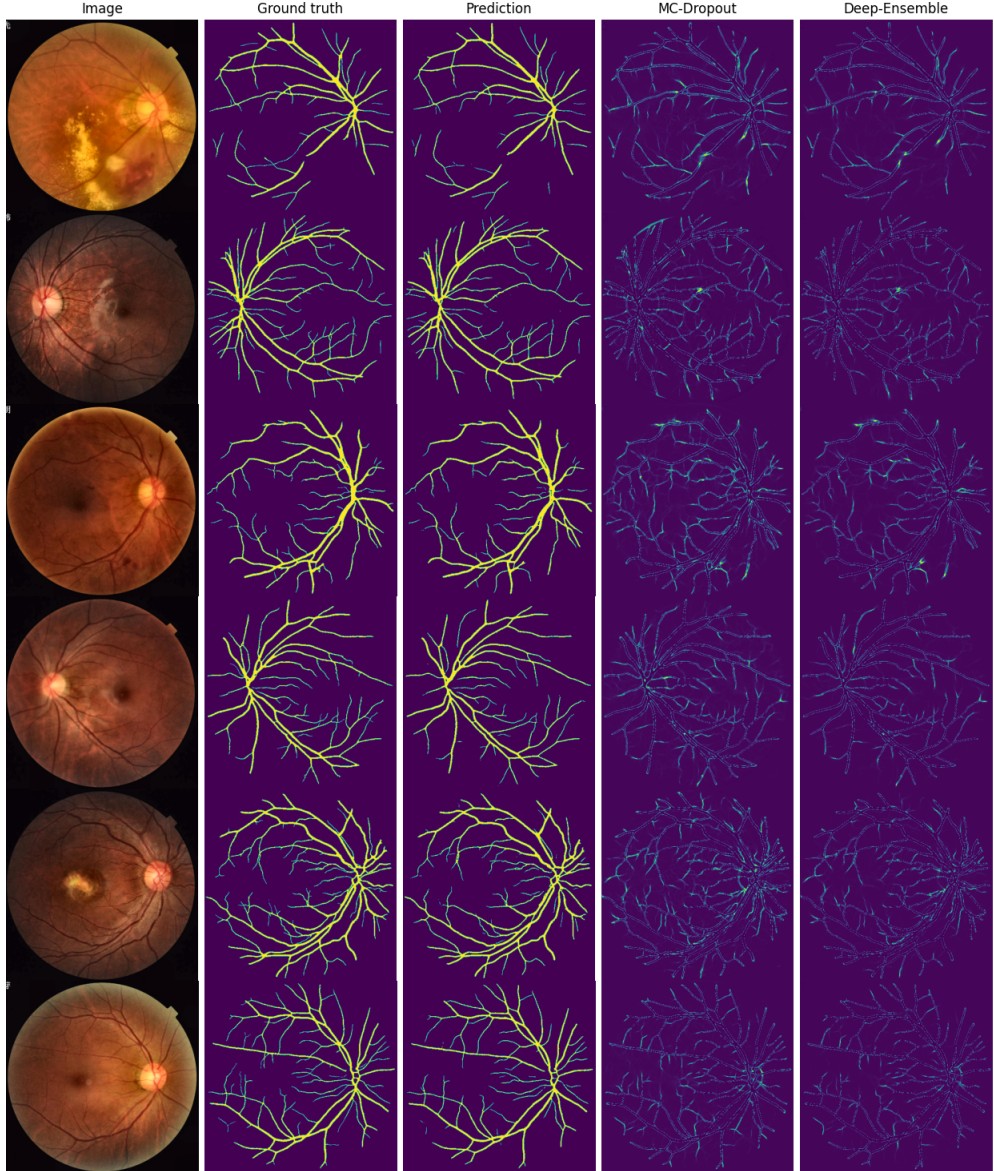

Figure 8: Qualitative comparison of pixelwise uncertainty estimates obtained from Deep Ensembles and MC Dropout. The uncertainties are structurally very similar. The MC Dropout uncertainties are computed from a single FR-Unet with dropout rate 0.1 at each convolutional layer. Note that the computational cost to determine the optimal dropout rate and position is very high, making it not substantially cheaper than the deep ensembles.

