# OpenReview forum: "Efficiently correcting patch-based segmentation errors to control image-level performance in retinal images"
_MIDL.io/2024/Conference — MIDL 2024 Oral_

### Official Review · Reviewer_fLiB · 2024-02-28

**Confidence:** 5
**Preliminary Rating:** 5
**Recommendation:** Poster
**Final Rating:** 5

**Summary:**

Authors propose a method to manually correct for segmentation results
with minimal effort. They first present a method for selecting patches
with potentially high segmentation error, then propose a way to
estimate how much DSC increase would one have by correcting the
patches. The first part is addressed by estimating segmentation
entropy at each pixel given an ensemble of segmentation results. The
second part is addressed by estimating DSC assuming calibrated
outputs. Experiments with retinal blood vessels on fundus images were
performed and consistent improvements over DSC with the correction
procedure were reported.

**Strengths:**

1. The paper is on a very relevant topic. Correction for segmentation
   is indeed an important topic for deployment.
2. The proposed procedure is novel to the best of my
   knowledge. Authors put together different components from the
   literature in a very elegant way and came up with a very practical
   solution to the problem.
3. The proposed procedure seems quite sound. Both the patch selection
   and DSC estimation are sound approaches. Of course, there are
   assumptions and especially the DSC estimation required correction
   as it did not fit to the assumption very well. Nonetheless, the
   correction seems to solve an issue.
4. Experimental analysis is quite thorough and authors evaluated the
   whole system as well as the DSC improvement separately. I really
   appreciated this. They also did a check with a baseline, that chose
   patches randomly from foreground pixels.
5. The article is very well written.

**Weaknesses:**

1. The title and abstract gives the impression that the proposed
   techniques are generally applicable to all medical image
   segmentation tasks. While this may still be possible, there is
   strong focus on a particular task and experiments only for that
   task are presented. Perhaps adjusting the title and the abstract
   may be useful.
2. Generation of the uncertainty maps is a computationally heavy
   mechanism. I am wondering whether one can get away by simply
   choosing areas with low contrast. Essentially, the uncertainty maps
   may be mostly highlighting low-contrast areas.

**Detailed Comments:**

I think the two points I mentioned as weaknesses can be discussed:
1. It would be useful to reconsider the title and the abstract, since all the experiments are presented only for one application.
2. Computational cost of running 5 segmentation models could be discussed and perhaps further ways to select patches may be discussed.

**Justification Of Final Rating:**

It is a very good article - and my opinion remained the same after the rebuttal.
This could be an oral presentation, however, I recommend a poster, since oral presentations should be reserved for perhaps more technically novel work.

**Justification Of The Preliminary Rating:**

There are many strengths and few weaknesses in this article. It is well written on a very relevant topic. The procedure is simply yet efficient. Methods are well justified and the entire model is well positioned in the literature.

**Questions To Address In The Rebuttal:**

I think the paper is quite good. I do not see a lot to discuss in the rebuttal.

**Special Issue:**

Yes

---

> ### Author Response · Authors · 2024-03-16
>
> Thank you for your helpful feedback and inputs.
>
> We have now adjusted the title and abstract to more clearly reflect the fact that the analyses in our paper were conducted on retinal vessel segmentation. The title now reads “Efficiently correcting patch-based segmentation errors to control image-level performance in retinal images”, and the abstract was slightly modified to emphasise the segmentation task. Please also note that we highlight our application area with the keywords “Retinal Blood Vessels” and “Fundus” below the abstract. We hope this avoids any further misunderstandings.
>
> The generation of the uncertainty maps is indeed computationally heavy during training, as it requires training multiple models. However, during test time, multiple forward passes are relatively cheap. Nevertheless, we agree in principle that it would be interesting to explore other strategies for candidate patch selection, either with different uncertainty techniques, or using low-level image statistics as the reviewer suggests. We have now added a brief discussion of the computational load of uncertainty estimation, and included the reviewer’s idea of selecting low contrast regions.

---

### Official Review · Reviewer_cqrS · 2024-03-01

**Confidence:** 4
**Preliminary Rating:** 4

**Summary:**

This paper addresses automatic segmentation uncertainty for retinal blood vessel segmentation from fundus images. The objective is to select a minimum number of smaller patches (e.g 81x81 in 512x512 images) that could benefit from "patch review", i.e. manual correction, a time intensive task.

To evaluate this aspect, the method is the following :
- From an 5-ensemble of a state of the art UNet retinal vessel segmentation model, a model-based pixel level uncertainty map is obtained (pixelwise entropy)
- It is observed that locally replacing the $k$ segmentation output patches having the highest average uncertainty with the ground truth improves segmentation quality. This is shown to be better (quite unsurprisingly) than randomly selecting patches (Fig. 2). A linear DSC gain is achieved with $k$

Another aspect of the paper is predicting increase in DSC performance without ground truth. An estimator of DSC proposed earlier (Li et al. 2022) is used to this end, which is shown to overestimate segmentation quality. This estimator is then calibrated to better reflect performances (Fig. 3).

The last aspect covers adapting the number of patches to be selected based on the estimator : i.e. selecting the lowest number $k$ that would beat a quality target.

**Strengths:**

First, hats off to the authors for the quality of the text and the figures : text is carefully written, figures are informative and of camera ready quality.
The paper proposes an ad hoc solution to consistently improve segmentation quality without reviewing the full image. This is an important topic insufficiently covered.
While the use of model based uncertainty could certainly be enriched from a growing literature on this topic, I find the proposed method simple, technically sound and elegant.

**Weaknesses:**

Other aspects could have been covered by the paper and should be mentioned at least in the discussion :
- The analysis of vessel like structures with the DSC is questionable, as global volumetric overlaps are typically biased towards higher volumes. There is a literature on better segmentation quality estimators, such as e.g. clDice (CVPR 2021) [1]
- The impact of the patch size is not discussed
- The impact of other epistemic uncertainty estimators beyond ensembling (e.g. MC dropout) is not discussed
- The assumption that a ground truth exists (and thus correction can be achieved by replacing with this perfect ground truth) means that there is no inter reader variability in annotating vessel structures, which I guess is not true even if I am not an expert in this field.

[1] https://github.com/jocpae/clDice

**Detailed Comments:**

Fig. 3 caption would read better with (a) by the model and (b) after manual review)

**Justification Of The Preliminary Rating:**

This is a very well written, sound paper that proposes an ad hoc solution to the improvement of vessel structure segmentation in retinal fundus image. While not particularly original in terms of methodology, this is a nice pipeline that could readily be implemented in clinical workflows. However, positioning with respect to other label noise / segmentation uncertainty literature is almost empty, which is I think the weaker spot of this work. I however recommend acceptance considering the solid work performed.

**Questions To Address In The Rebuttal:**

- limitations of the Dice score
- existence of other epistemic uncertainty measures beyond entropy of ensembles
- mention inter reader variability and limitation of the study in this regard

---

> ### Author Response · Authors · 2024-03-16
>
> Thank you for taking the time to read our work carefully and providing constructive as well as  helpful feedback.
>
> We agree that clDSC is a particularly useful metric for the task of vessel segmentation, as has also been suggested by reviewer N9Um. We now included clDSC and the limitations of DSC  in our discussion, and are planning to implement a clDSC estimator as an extension of this work. Nevertheless, we believe our work using DSC estimation can be a useful starting point for many, as it is generally applicable in many segmentation tasks beyond tube-like structures.
>
> We agree with the reviewer that a discussion of other uncertainty estimators will strengthen the paper. We used entropy of deep ensembles as one example technique, but of course, our modular framework allows replacing this step with any other approach for generating uncertainty maps. As suggested by reviewer fLiB, one could even consider candidate patch selection based on image statistics alone. We have now included a discussion of different approaches to uncertainty estimation. In addition, we have also qualitatively compared the uncertainty maps produced by our ensemble with MC dropout-generated uncertainty maps (added in App. F). Both methods resulted in very similar uncertainty maps and we would not expect the experimental results to change substantially in this case.
>
> For simplicity, we assumed a perfect oracle, but we agree that a perfect groundtruth without inter-rater variability may be unrealistic. However, we hypothesise that the expert performance may be fairly high and consistent in an error correction framework where attention is drawn to small and few areas. However, as a proof of concept it could for example be interesting to evaluate our method with multi-annotator data, where the oracle patch would then be provided by a different annotator than the reference annotation. Another interesting future avenue would be to actually measure and report manual performance on patches in an error correction setting. We have now included a discussion of inter-rater variability and the limitation of our evaluation in this regard.
>
> Finally, the reviewer has commented on the choice of patch size in our pipeline. We have heuristically chosen a patch size of 81 as a tradeoff between annotation effort and field of view: we assumed that it may be useful to have a large enough patch to include some contextual information, while keeping the area as small as possible to reduce the manual annotation burden. We have now included an additional analysis (App. D) where we address the question “Is it better to review multiple smaller patches or fewer larger patches?”. We compared the increase in DSC when using different patch sizes, where we fixed the total size of the reviewed area. That is, we compared the effect of reviewing 16 patches of size 41x41, 4 patches of size 81x81, and 1 patch of size 161x161. We found that reviewing 16 patches of size 41x41 had a similar effect as reviewing 4 patches of size 81x81, while reviewing only a single large patch unsurprisingly yielded smaller DSC improvements.

---

### Official Review · Reviewer_N9Um · 2024-03-03

**Confidence:** 5
**Preliminary Rating:** 5
**Recommendation:** Oral
**Final Rating:** 5

**Summary:**

This paper proposes a strategy to correct segmentations in a smart way so that the dice score increases as quick as possible. For this, an ensemble of methods is used to generate a pixelwise uncertainty map, and this is then used to source patches that are potentially wrong. Among those patches, the authors are able to estimate how much would the global dice score increase if someone corrected each of them (this idea is brilliant, I think) . Therefore in test time, the model can extract the patches that would be more efficient to correct to achieve a desired quality.

**Strengths:**

I like a lot the paper. It is original in that I have not seen before someone proposing to find the local areas of the images that could be reviewed to improve a segmentation with a prescribed overlap quality in mind. It is also technically rigorous, and the experiments are methodic and details are well-specified, and make sense. This one should be accepted for sure, in my opinion.

**Weaknesses:**

- My main question is, how relevant is the correction of a few dice score points to downstream clinical applications? I am thinking of a "bad" vessel segmentation in a fundus images, and I am not sure that I would want to improve its dice, becuase that could lead me to spend time making some bigvessels a bit thicker or thinner, according to the preferences of a particular annotator. But in the end, what matters most is to get the vessel tree right, without breaking vessels, don't you think? Let us discuss this in rebuttal, I am willing to be convinced otherwise. Also, please see below a suggestion.

- Not a great concern, but I wonder if training on 512x512 for images that are 4x that size would not already lose vessels during downsampling? Also, does all the work happen on the downsampled images, or are they only downsampled for training but then the method works at native resoution, as it should? I mean, do you measure dice at native resolution upsampling the result of the model, or do you stay at 512x512 from beginning to end.

**Detailed Comments:**

- My main suggestion is, couldn't we think of using the centerline dice metric (https://doi.org/10.1109/CVPR46437.2021.01629) instead of the dice score? In essence it measure the part of the skeleton of the prediction that is not inside the ground-truth plus the part of the skeleton of the ground-truth that is not in the prediction. Well, and it has a differentiable version you can use for training, but that is not relevant here I think. The kind of solutions favoured by this metric enforces topological consistency, and could be more useful in a clinical setting, maybe?

- The FIVES dataset contains four different classes of images, namely non-pathological, glaucomoatous, with DR and with AMD. I wonder how would the performance of this strategy vary when testing in different subgroups? I mean, the authors are among the most qualified experts to answer this question, given their recent work :)

- Can I please ask the authors to have a look at this MICCAI 2018? It basically did the same as Robinson et al.'s MICCAI 2018, but with retinal vessel segmentation, so one would think it is even more related to this paper. The authors chose to use a title that did not include "segmentation quality prediction" but "a no-reference quality metric for vessel segmentation", which has resulted in little attention drawn to it, but I still think it is relevant enough to be discussed in the related work section.

**Justification Of Final Rating:**

Nothing to add here, I had already given this one a Strong Accept before the rebuttal, and if any, the paper has improved. The paper is indeed strong, and what is better, it has extremely interesting directions to expand, which warrants a very nice future extended work (the MELBA special issue, maybe?). Anyway, a pleasure to read.

**Justification Of The Preliminary Rating:**

There are very little things I would modify from this paper, I would just recommend an acceptance "as is". Apparently openreview requires me to write a minimum of 200 characters, but I don't think I need them.

**Questions To Address In The Rebuttal:**

Nothing in particular, maybe clarify my doubts above?

**Special Issue:**

Yes

---

> ### Comment · Reviewer_N9Um · 2024-03-12
> **Forgot to a link to a reference I mentioned**
>
> Sorry, while re-reading my own review I noticed that in the third detailed comment, I forgot to add the actual link to the reference I mentioned, here it is: https://doi.org/10.1007/978-3-030-00928-1_10

---

> > ### Author Response · Authors · 2024-03-15
> >
> > Dear reviewer, thanks for the link! We had searched for it and assumed that was the paper you were referring to. We will post our response to your review shortly, and look forward to the discussion phase.

---

> ### Author Response · Authors · 2024-03-16
> **Full Response**
>
> Thank you for taking the time to read our paper in depth and for your thoughtful feedback.
>
> We believe applying our  method to DSC estimation is a useful starting point with general applicability in many segmentation tasks. However, we agree that clDSC is a particularly useful metric for the task of vessel segmentation and have now included a discussion of clDSC and the limitations of DSC in our paper. We have now also started to investigate how we could extend our estimator to clDSC for an extension of this work.
>
> We agree that subgroup analysis is relevant and interesting here. We have now evaluated our method  in the individual subgroups and found similar results as in our original analysis. The subgroup with the lowest segmentation performance (Glaucoma) profited most from the patch review. Furthermore, we found that the pathological subgroups contained individual images where the review was particularly helpful, whereas such “outliers” did not exist for the normal subgroup. We have added these supplementary results to the appendix.
>
> Regarding your comment about image resolution: We indeed carried out all our experiments on a 512x512 resolution. In principle, the method could also work in native resolution, using upsampled predictions and uncertainty maps, and inserting high-resolution oracle data. However, we now found that simply upsampling the results of our segmentation model to native resolution led to a DSC drop by 0.05, indicating that indeed, some information was lost in the original downsampling step. Therefore we feel that conceptually, it is more meaningful to evaluate our method in 512x512 resolution to not confound the effects of downsampling with the effects of our method. We welcome further comments on this matter, and also believe this would actually raise an interesting follow-up question for future extension: Is it useful to correct lower-resolution models with high-resolution manual reviews?
>
> Thank you for pointing us to the MICCAI’18 paper, it is indeed very related. We have included it as related work in the introduction.

---

> > ### Comment · Reviewer_N9Um · 2024-03-21
> >
> > Oh, that is a great idea! If I was able to train a model at low resolution, and then (learn to) do the corrections at high-res, patch-wise, that would be a nice thing to have. You see, a problem I almost always encounter (with high-res or 3d imaging) is that I am forced to train with patches in order to get good quality solutions. But when training with patches the "big picture" is lost to the model. I a recent project I have a large scan with an object always in the middle, I cannot feed the entire scan to the model due to OOM in the gpu, so I have to train patchwise, and then the prior info that there is a single object in the middle is lost, which is very meh.
> >
> > Looking forward to read about your estimator of the clDice score, it would be a really useful tool to have. Also thanks for paying attention to my comments, even though you had no chance that I would increase my rating :)

---

### Author Response · Authors · 2024-03-16
**Version with highlighted changes**

Dear reviewers, thank you very much for your thorough and helpful feedback. We are very happy that you appreciated our work and its presentation. You have raised relevant and interesting questions and concerns which we believe have improved our paper.

We have submitted a revised version of the manuscript. In addition, we’ve uploaded marked-up version on google drive (https://drive.google.com/file/d/1ZGP9o4nQJqZZwvS9B8c4utDyVxC4cQ9Y/view?usp=sharing), where new material is marked in red. Please note that we also made a few minor modifications to keep the paper within the page limit after incorporating the reviewer comments.

We have addressed your comments in the individual posts below and look forward to a fruitful discussion.

---

### Author Response · Authors · 2024-03-27
**Revised Fig. 2**

Dear all,

during our internal review, we have identified a minor bug in the code that displays the results of the baseline method (random patch selection) for Fig. 2 c, d.
Instead of showing the baseline performance for the respective subgroups (well and poorly segmented images), we've accidentally shown the baseline performance for the entire test set in all panels b, c and d.

We have fixed that and find, as expected, that the baseline performance is higher for poorly segmented images than across the entire test set (panel d) and lower for the well segmented images.
However, this does neither affect the overall result nor the interpretation.

We have uploaded a revised version of the manuscript with the corrected figure panels.

---

### Meta-Review · Area_Chair_VnC8 · 2024-04-04

**Recommendation:** Accept (Oral)
**Confidence:** 5

**Metareview:**

This paper proposes a strategy to correct segmentations in a smart way so that the dice score increases as quick as possible. This paper is orginal and well-written. All reviewers agreed to accept it.

---

### Decision · Program_Chairs · 2024-04-05

Accept (Oral)